# Pymablock: an algorithm and a package for quasi-degenerate perturbation theory

Isidora Araya Day[1,2]⋆ , Sebastian Miles[1], Hugo K. Kerstens[2] , Daniel Varjas[3,4] ,
Anton R. Akhmerov[2†]

**1** QuTech, Delft University of Technology, Delft 2600 GA, The Netherlands
**2** Kavli Institute of Nanoscience, Delft University of Technology, 2600 GA Delft, The Netherlands
**3** Max Planck Institute for the Physics of Complex Systems, Nöthnitzer Strasse 38, 01187 Dresden, Germany
**4** Institute for Theoretical Solid State Physics, IFW Dresden and Würzburg-Dresden Cluster of Excellence ct.qmat, Helmholtzstr. 20, 01069 Dresden, Germany
⋆ iarayaday@gmail.com † pymablock@antonakhmerov.org

April 4, 2024

## Abstract

A common technique in the study of complex quantum-mechanical systems is to reduce the number of degrees of freedom in the Hamiltonian by using quasi-degenerate perturbation theory. While the Schrieffer–Wolff transformation achieves this and constructs an effective Hamiltonian, its scaling is suboptimal, and implementing it efficiently is both challenging and error-prone. We introduce an algorithm for constructing an equivalent effective Hamiltonian as well as a Python package, Pymablock, that implements it. Our algorithm combines an optimal asymptotic scaling with a range of other improvements. The package supports numerical and analytical calculations of any order and it is designed to be interoperable with any other packages for specifying the Hamiltonian. We demonstrate how the package handles constructing a k.p model, analyses a superconducting qubit, and computes the low-energy spectrum of a large tight-binding model. We also compare its performance with reference calculations and demonstrate its efficiency.

# 1  Introduction

Effective models enable the study of complex quantum systems by reducing the dimensionality of the Hilbert space. Their construction separates the low and high-energy subspaces by block-diagonalizing a perturbed Hamiltonian

$$\mathcal{H} = \begin{pmatrix} H_0^{AA} & 0 \\ 0 & H_0^{BB} \end{pmatrix} + \mathcal{H}', \tag{1}$$

where $H_0^{AA}$ and $H_0^{BB}$ are separated by an energy gap, and $\mathcal{H}'$ is a series in a perturbative parameter. This procedure requires finding a series of the basis transformation $\mathcal{U}$ that is unitary and that also cancels the off-diagonal block of the transformed Hamiltonian order by order, as shown in Fig. 1. The low-energy effective Hamiltonian $\tilde{\mathcal{H}}^{AA}$ is then a series in the perturbative parameter, whose eigenvalues and eigenvectors are approximate solutions of the complete Hamiltonian. As a consequence, the effective model is sufficient to describe the low-energy properties of the original system while also being simpler and easier to handle.

A common approach to constructing an effective Hamiltonian is the Schrieffer–Wolff transformation [1, 2], also known as Löwdin partitioning [3], or quasi-degenerate perturbation theory. This method parameterizes the unitary transformation $\mathcal{U} = e^{-\mathcal{S}}$ and finds the series $\mathcal{S}$ that decouples the $A$ and $B$ subspaces of $\tilde{\mathcal{H}} = e^{\mathcal{S}} \mathcal{H} e^{-\mathcal{S}}$. This idea enabled advances in multiple fields of quantum physics. As an example, all the k.p models are a result of treating crystalline momentum as a perturbation that only weakly mixes atomic orbitals separated in energy [4]. More broadly, this method serves as a go-to tool in the study of superconducting circuits and quantum dots, where couplings between circuit elements and drives are treated as perturbations to reproduce the dynamics of the system [5, 6].

Constructing effective Hamiltonians is, however, both algorithmically complex and computationally expensive. This is a consequence of the recursive equations that define the unitary transformation, which require an exponentially growing number of matrix products in each order. In particular, already a 4-th order perturbative expansion that is necessary for many applications may require hundreds of terms. While the computational complexity is only a nuisance when analysing model systems, it becomes a bottleneck whenever the Hilbert space is high-dimensional. Several alternative approaches improve the performance of the Schrieffer–Wolff algorithm by either using different parametrizations of the unitary transformation [3, 7–10], adjusting the problem setting to density matrix perturbation theory [11, 12], or a finding a similarity transform instead of a unitary [13]. A more recent line of research even applies the ideas of Schrieffer–Wolff transformation to quantum algorithms for the study of many-body systems [14, 15]. Despite

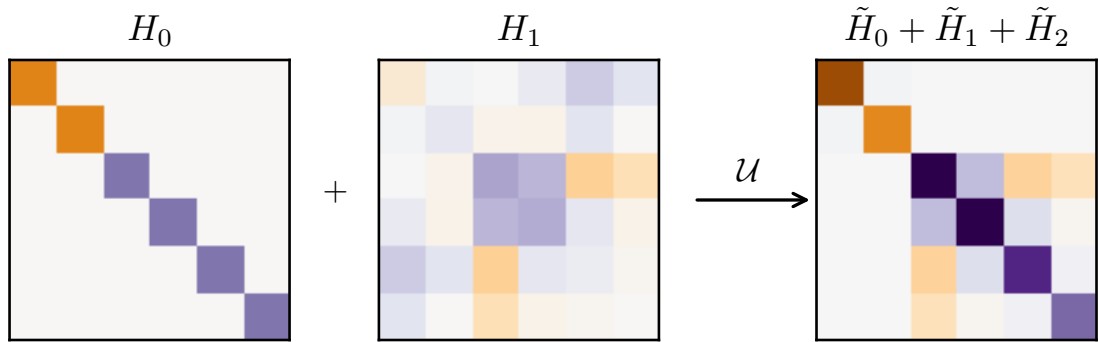

Figure 1: Block-diagonalization of a Hamiltonian with a first order perturbation.

these advances, neither of the approaches combines an optimal scaling with the ability to construct effective Hamiltonians.

We introduce an algorithm to construct effective models with optimal scaling, thus making it possible to find high order effective models for systems with millions of degrees of freedom. This algorithm exploits the efficiency of recursive evaluations of series satisfying polynomial constraints and obtains the same effective Hamiltonian as the Schrieffer–Wolff transformation. We make the algorithm available via the open source package Pymablock [1](PYthon MAtrix BLOCK-diagonalization), a versatile tool for the study of numerical and symbolic models.

## 2   Constructing an effective model

We illustrate the construction of effective models by considering several representative examples. The simplest application of effective models is the reduction of finite symbolic Hamiltonians, which appear in the derivation of low-energy dispersions of materials. Starting from a tight-binding model, one performs Taylor expansions of the Hamiltonian near a $k$-point, and then eliminates several high-energy states [4, 16]. In the study of superconducting qubits, for example, the Hamiltonian contains several bosonic operators, so its Hilbert space is infinite-dimensional, and the coupling between bosons makes the Hamiltonian impossible to diagonalize. The effective qubit model describes the analytical dependence of qubit frequencies and couplings on the circuit parameters [5, 17–21]. This allows to design circuits that realize a desired qubit Hamiltonian, as well as ways to understand and predict qubit dynamics, for which computational tools are being actively developed [22–24]. Finally, mesoscopic quantum devices are described by a single particle tight-binding model with short range hoppings. This produces a numerical Hamiltonian that is both big and sparse, which allows to compute a few of its states but not the full spectrum [25]. Because only the low-energy states contribute to observable properties, deriving how they couple enables a more efficient simulation of the system's behavior.

Pymablock treats all the problems, including the ones above, using a unified approach that only requires three steps:

- Define a Hamiltonian

- Call `pymablock.block_diagonalize`

- Request the desired order of the effective Hamiltonian

---

[1]The documentation and tutorials are available in https://pymablock.readthedocs.io/

The following code snippet shows how to use Pymablock to compute the fourth order correction to an effective Hamiltonian $\tilde{\mathcal{H}}$:

```
# Define perturbation theory
H_tilde, *_ = block_diagonalize([H_0, H_1], subspace_eigenvectors=[vecs_A, vecs_B])

# Request 4th order correction to the effective Hamiltonian
H_AA_4 = H_tilde[0, 0, 4]
```

The function `block_diagonalize` interprets the Hamiltonian $H_0 + H_1$ as a series with two terms, zeroth and first order and calls the block diagonalization routine. The subspaces to decouple are spanned by the eigenvectors `vecs_A` and `vecs_B` of $H_0$. This is the main function of Pymablock, and it is the only one that the user ever needs to call. Its first output is a multivariate series whose terms are different blocks and orders of the transformed Hamiltonian. Calling `block_diagonalize` only defines the computational problem, whereas querying the elements of `H_tilde` does the actual calculation of the desired order. This interface treats arbitrary formats of Hamiltonians and system descriptions on the same footing and supports both numerical and symbolic computations.

## 2.1 k.p model of bilayer graphene

To illustrate how to use Pymablock with analytic models, we consider two layers of graphene stacked on top of each other, as shown in Fig. 2. Our goal is to find the low-energy model near the **K** point [16]. To do this, we first construct the tight-binding model Hamiltonian of bilayer graphene. The main features of the model are its 4-atom unit cell

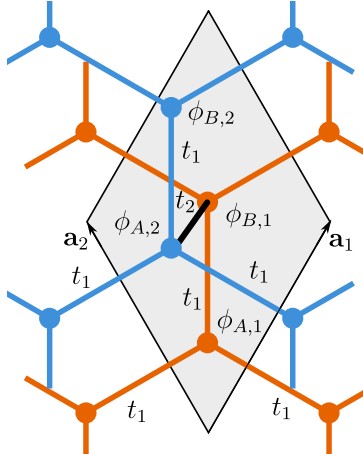

Figure 2: Crystal structure and hoppings of AB-stacked bilayer graphene.

spanned by vectors $\mathbf{a}_1 = (1/2, \sqrt{3}/2)$ and $\mathbf{a}_2 = (-1/2, \sqrt{3}/2)$, and with wave functions $\phi_{A,1}, \phi_{B,1}, \phi_{A,2}, \phi_{B,2}$, where $A$ and $B$ indices are the two sublattices, and $1, 2$ are the layers. The model has hoppings $t_1$ and $t_2$ within and between the layers, respectively, as shown in Fig. 2. We also include a layer-dependent onsite potential $\pm m$.

We define the Bloch Hamiltonian using the Sympy package for symbolic Python [26].

```
t_1, t_2, m = sympy.symbols("t_1 t_2 m", real=True)
alpha = sympy.symbols(r"\alpha")

H = Matrix([
    [m, t_1 * alpha, 0, 0],
    [t_1 * alpha.conjugate(), m, t_2, 0],
```

```
    [0, t_2, -m, t_1 * alpha],
    [0, 0, t_1 * alpha.conjugate(), -m]]
)
```

$$H = \begin{pmatrix} m & t_1\alpha & 0 & 0 \\ t_1\alpha^* & m & t_2 & 0 \\ 0 & t_2 & -m & t_1\alpha \\ 0 & 0 & t_1\alpha^* & -m \end{pmatrix}$$

where $\alpha(\mathbf{k}) = 1 + e^{i\mathbf{k}\cdot\mathbf{a_1}} + e^{\mathbf{k}\cdot\mathbf{a_2}}$, with $k$ the wave vector. We consider $\mathbf{K} = (4\pi/3, 0)$ the reference point point in $\mathbf{k}$-space: $\mathbf{k} = (4\pi/3 + k_x, k_y)$ because $\alpha(\mathbf{K}) = 0$, making $k_x$ and $k_y$ small perturbations. Additionally, we consider $m \ll t_2$ a perturbative parameter.

To call `block_diagonalize`, we need to define the subspaces for the block diagonalization, so we compute the eigenvectors of the unperturbed Hamiltonian at the $\mathbf{K}$ point, $H(\alpha(\mathbf{K}) = m = 0)$. Then, we substitute $\alpha(\mathbf{k})$ into the Hamiltonian, and call the block diagonalization routine using that $k_x$, $k_y$, and $m$ are perturbative parameters via the `symbols` argument.

```
vecs = H.subs({alpha: 0, m: 0}).diagonalize(normalize=True)[0]

H_tilde, U, U_adjoint = block_diagonalize(
    H.subs({alpha: alpha_k}),
    symbols=(k_x, k_y, m),
    subspace_eigenvectors=[vecs[:, :2], vecs[:, 2:]] # AA, BB
)
```

The order of the variables in the perturbative series will be that of `symbols`. For example, requesting the term $\propto k_x^i k_y^j m^l$ from the effective model is done by calling `H_tilde[0, 0, i, j, l]`, where the first two indices are the block indices (AA). The series of the unitary transformation $U$ and $U^\dagger$ are also defined, and we may use them to transform other operators.

We collect corrections up to third order in momentum to compute the standard quadratic dispersion of bilayer graphene and trigonal warping. We query these terms from `H_tilde` and those proportional to mass to obtain the effective Hamiltonian (shown as produced by the code)[2]:

$$\tilde{H}_{\text{eff}} = \begin{bmatrix} m & \frac{3t_1^2}{4t_2}(-k_x^2 - 2ik_xk_y + k_y^2) \\ \frac{3t_1^2}{4t_2}(-k_x^2 + 2ik_xk_y + k_y^2) & -m \end{bmatrix} +$$
$$\begin{bmatrix} \frac{3mt_1^2}{2t_2^2}(-k_x^2 - k_y^2) & \frac{\sqrt{3}t_1^2}{8t_2}(k_x^3 - 5ik_x^2k_y + 9k_xk_y^2 + 3ik_y^3) \\ \frac{\sqrt{3}t_1^2}{8t_2}(k_x^3 + 5ik_x^2k_y + 9k_xk_y^2 - 3ik_y^3) & \frac{3mt_1^2}{2t_2^2}(k_x^2 + k_y^2) \end{bmatrix}$$

The first term is the standard quadratic dispersion of gapped bilayer graphene. The second term contains trigonal warping and the coupling between the gap and momentum. All the terms take less than two seconds in a personal computer to compute.

## 2.2 Dispersive shift of a transmon qubit coupled to a resonator

The need for analytical effective Hamiltonians often arises in circuit quantum electrodynamics (cQED) problems, which we illustrate by studying a transmon qubit coupled to a resonator [5]. Specifically, we choose the standard problem of finding the frequency shift of the resonator due to its coupling to the qubit, a phenomenon used to measure the qubit's

---

[2]The full code is available at https://pymablock.readthedocs.io/en/latest/tutorial/bilayer_graphene.html.

state [17]. The Hamiltonian of the system is given by

$$\mathcal{H} = -\omega_t(a_t^\dagger a_t - \frac{1}{2}) + \frac{\alpha}{2}a_t^\dagger a_t^\dagger a_t a_t + \omega_r(a_r^\dagger a_r + \frac{1}{2}) - g(a_t^\dagger - a_t)(a_r^\dagger - a_r), \tag{2}$$

where $a_t$ and $a_r$ are bosonic annihilation operators of the transmon and resonator, respectively, and $\omega_t$ and $\omega_r$ are their frequencies. The transmon has an anharmonicity $\alpha$, so that its energy levels are not equally spaced. In presence of both the coupling $g$ between the transmon and the resonator and the anharmonicity, this Hamiltonian admits no analytical solution. We therefore treat $g$ as a perturbative parameter.

To deal with the infinite dimensional Hilbert space, we observe that the perturbation only changes the occupation numbers of the transmon and the resonator by $\pm 1$. Therefore computing $n$-th order corrections to the $n_0$-th state allows to disregard states with any occupation numbers larger than $n_0 + n/2$. We want to compute the second order correction to the levels with occupation numbers of either the transmon or the resonator being 0 and 1. We accordingly truncate the Hilbert space to the lowest 3 levels of the transmon and the resonator. The resulting Hamiltonian is a $9 \times 9$ matrix that we construct using Sympy [26].

Finally, to compute the energy corrections of the lowest levels, we call `block_diagonalize` for each state separately, replicating a regular perturbation theory calculation for single wavefunctions. To do this, we observe that $H_0$ is diagonal, and use `subspace_indices` to assign the elements of its eigenbasis to the $A$ (0) or $B$ (1) subspace. For example, to find the qubit-dependent frequency shift of the resonator, $\chi$, we start by computing the second order correction to $|0_t 0_r\rangle$:

```
indices = [0, 1, 1, 1, 1, 1, 1, 1, 1] # 00 is the first state in the basis
H_tilde, *_ = block_diagonalize(H, subspace_indices=indices, symbols=[g])
H_tilde[0, 0, 2][0, 0] # 2nd order correction
```

$$E_{00}^{(2)} = \frac{g^2}{-\omega_r + \omega_t}. \tag{3}$$

Repeating this process for the other levels requires changing `subspace_indices` according to the basis of $H$, and yields the desired resonator frequency shift:

$$\begin{aligned}
\chi &= (E_{11}^{(2)} - E_{10}^{(2)}) - (E_{01}^{(2)} - E_{00}^{(2)}) \\
&= -\frac{2g^2}{\alpha + \omega_r - \omega_t} + \frac{2g^2}{-\alpha + \omega_r + \omega_t} - \frac{2g^2}{\omega_r + \omega_t} + \frac{2g^2}{\omega_r - \omega_t} \\
&= -\frac{4\alpha g^2 \left(\alpha\omega_t - \omega_r^2 - \omega_t^2\right)}{(\omega_r - \omega_t)(\omega_r + \omega_t)(-\alpha + \omega_r + \omega_t)(\alpha + \omega_r - \omega_t)}.
\end{aligned} \tag{4}$$

In this example, we have not used the rotating wave approximation, including the frequently omitted counter-rotating terms $\sim a_r a_t$ to illustrate the extensibility of Pymablock. Computing higher order corrections to the qubit frequency only requires increasing the size of the truncated Hilbert space and requesting `H_tilde[0, 0, n]` to the desired order $n$.

## 2.3 Induced gap in a double quantum dot

Large systems pose an additional challenge due to the cubic scaling of linear algebra routines with matrix size. To overcome this, Pymablock is equipped with an implicit method, which utilizes the sparsity of the input and avoids the construction of the full transformed Hamiltonian. We illustrate the efficiency of this method by applying it to a

system of two quantum dots coupled to a superconductor between them, shown in Fig. 3, and described by the Bogoliubov-de Gennes Hamiltonian:

$$H_{BdG} = \begin{cases} (\mathbf{k}^2/2m - \mu_{sc})\sigma_z + \Delta\sigma_x & \text{for } L/3 \leq x \leq 2L/3, \\ (\mathbf{k}^2/2m - \mu_n)\sigma_z & \text{otherwise,} \end{cases} \tag{5}$$

where the Pauli matrices $\sigma_z$ and $\sigma_x$ act in the electron-hole space, $\mathbf{k}$ is the 2D wave vector, $m$ is the effective mass, and $\Delta$ the superconducting gap.

We use the Kwant package [27] to build the Hamiltonian of the system [3], which we define over a square lattice of $L \times W = 200 \times 40$ sites. On top of this, we consider two perturbations: the barrier strength between the quantum dots and the superconductor, $t_b$, and an asymmetry of the dots' potentials, $\delta\mu$.

The system is large: it is a sparse array of size $63042 \times 63042$, with $333680$ non-zero elements, so even storing all the eigenvectors would take 60 GB of memory. The perturbations are also sparse, with 632, and 126084 non-zero elements for the barrier strength and the potential asymmetry, respectively. The sparsity structure of the Hamiltonian and the perturbations is shown in the left panel of Fig. 3, where we use a smaller system of $L \times W = 8 \times 2$ for visualization. Therefore, we use sparse diagonalization [28] and compute only four eigenvectors of the unperturbed Hamiltonian closest to zero energy, which are the Andreev states of the quantum dots.

```
vals, vecs = scipy.sparse.linalg.eigsh(h_0, k=4, sigma=0)
vecs, _ = scipy.linalg.qr(vecs, mode="economic")  # orthogonalize the vectors
```

We now call the block diagonalization routine and provide the computed eigenvectors.

```
H_tilde, *_ = block_diagonalize([h_0, barrier, dmu], subspace_eigenvectors=[vecs])
```

Because we only provide the low-energy subspace, Pymablock uses the implicit method. Calling `block_diagonalize` is now the most time-consuming step because it requires pre-computing several decompositions of the full Hamiltonian. It is, however, manageable and it only produces a constant overhead of less than three seconds.

To compute the spectrum, we collect the lowest three orders in each parameter in an appropriately sized tensor.

```
h_tilde = np.array(np.ma.filled(H_tilde[0, 0, :3, :3], fill_value).tolist())
```

This takes two more seconds to run, and we can now compute the low-energy spectrum after rescaling the perturbative corrections by the magnitude of each perturbation.

```
def effective_energies(h_tilde, barrier, dmu):
    barrier_powers = barrier ** np.arange(3).reshape(-1, 1, 1, 1)
    dmu_powers = dmu ** np.arange(3).reshape(1, -1, 1, 1)
    return scipy.linalg.eigvalsh(
        np.sum(h_tilde * barrier_powers * dmu_powers, axis=(0, 1))
    )
```

Finally, we plot the spectrum of the 2 Andreev states in Fig. 3. As expected, the crossing at $E = 0$ due to the dot asymmetry is lifted when the dots are coupled to the superconductor. In addition, we observe how the proximity gap of the dots increases with the coupling strength.

---

[3]The full code is available at https://pymablock.readthedocs.io/en/latest/tutorial/induced_gap.html.

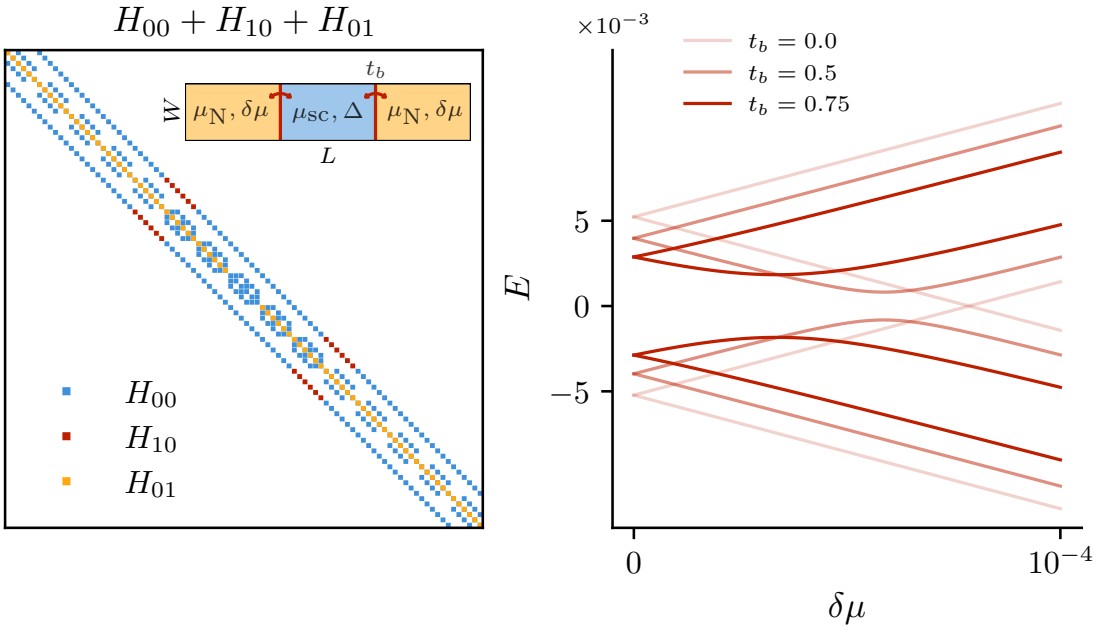

Figure 3: Hamiltonian (left) and Andreev levels (right) of two quantum dots coupled to a superconductor (inset). The barrier $t_b$ between the dots and the superconductor, $H_{10}$, and the asymmetry $\delta\mu$ between the dots' potential, $H_{01}$, are perturbations.

Computing the spectrum of the system for 3 points in parameter space would require the same time as the total runtime of Pymablock in this example. This demonstrates the speed of the implicit method and the efficiency of Pymablock's algorithm.

## 3  Perturbative block-diagonalization algorithm

### 3.1  Problem statement

Pymablock finds a series of the unitary transformation $\mathcal{U}$ (we use calligraphic letters to denote series) that block-diagonalizes the Hamiltonian

$$\mathcal{H} = H_0 + \mathcal{H}', \quad H_0 = \begin{pmatrix} H_0^{AA} & 0 \\ 0 & H_0^{BB} \end{pmatrix}, \tag{6}$$

with $\mathcal{H}' = \mathcal{H}'_D + \mathcal{H}'_O$ containing an arbitrary number and orders of perturbations with block-diagonal and block-offdiagonal components, respectively. The series here may be multivariate, and they represent sums of the form

$$\mathcal{A} = \sum_{n_1=0}^{\infty} \sum_{n_2=0}^{\infty} \cdots \sum_{n_k=0}^{\infty} \lambda_1^{n_1} \lambda_2^{n_2} \cdots \lambda_k^{n_k} A_{n_1, n_2, \ldots, n_k}, \tag{7}$$

where $\lambda_i$ are the perturbation parameters and $A_{n_1, n_2, \ldots, n_k}$ are linear operators. The problem statement, therefore, is finding $\mathcal{U}$ and $\tilde{\mathcal{H}}$ such that

$$\tilde{\mathcal{H}} = \mathcal{U}^\dagger \mathcal{H} \mathcal{U}, \quad \tilde{\mathcal{H}}^{AB} = 0, \quad \mathcal{U}^\dagger \mathcal{U} = 1, \tag{8}$$

which is schematically shown in Fig. 1. Series multiply according to the Cauchy product:

$$\mathcal{C} = \mathcal{A}\mathcal{B} \Leftrightarrow C_{\mathbf{n}} = \sum_{\mathbf{m}+\mathbf{p}=\mathbf{n}} A_{\mathbf{m}} B_{\mathbf{p}}.$$

The Cauchy product is the most expensive operation in perturbation theory, because it involves a large number of multiplications between potentially large matrices. For example, evaluating $\mathbf{n}$-th order of $\mathcal{C}$ requires $\sim \prod_i n_i \equiv N$ multiplications of the series elements.[4] A direct computation of all the possible index combinations in a product between three series $\mathcal{A}\mathcal{B}\mathcal{C}$ would have a higher cost $\sim N^2$, however, if we use associativity of the product and compute this as $(\mathcal{A}\mathcal{B})\mathcal{C}$, then the scaling of the cost stays $\sim N$.

There are many ways to solve the problem (8) that give identical expressions for $\mathcal{U}$ and $\tilde{\mathcal{H}}$. We are searching for a procedure that satisfies two additional constraints:

- It has the same complexity scaling as a Cauchy product, and therefore $\sim N$ multiplications per additional order.

- It does not require multiplications by $H_0$.

- It requires only one Cauchy product by $\mathcal{H}_D$, the block-diagonal part of $\mathcal{H}$.

The first requirement is that the algorithm scaling is optimal: the desired expression at least contains a Cauchy product of $\mathcal{U}$ and $\mathcal{H}$. Therefore the complexity scaling of the complete algorithm may not become lower than the complexity of a Cauchy product and we aim to reach this lower bound. The second requirement is because in perturbation theory, $n$-th order corrections to $\tilde{\mathcal{H}}$ carry $n$ energy denominators $1/(E_i - E_j)$, where $E_i$ and $E_j$ are the eigenvalues of $H_0$ belonging to different subspaces. Therefore, any additional multiplications by $H_0$ must cancel with additional energy denominators. Multiplying by $H_0$ is therefore unnecessary work, and it gives longer intermediate expressions. The third requirement we impose by considering a case in which $\mathcal{H}_O = 0$, where $\mathcal{H}_D$ must at least enter $\tilde{\mathcal{H}}$ as an added term, without any products. Moreover, because $\mathcal{U}$ depends on the entire Hamiltonian, there must be at least one Cauchy product by $\mathcal{H}'_D$. The goal of our algorithm is thus to be efficient and to produce compact results that do not require further simplifications.

## 3.2    Existing solutions

A common approach to constructing effective Hamiltonians is to use the Schrieffer–Wolff transformation [1]:

$$\tilde{\mathcal{H}} = e^{\mathcal{S}}\mathcal{H}e^{-\mathcal{S}}, e^{\mathcal{S}} = 1 + \mathcal{S} + \frac{1}{2!}\mathcal{S}\mathcal{S} + \frac{1}{3!}\mathcal{S}\mathcal{S}\mathcal{S} + \cdots, \tag{9}$$

where $\mathcal{S} = \sum_n S_n$ is an antihermitian polynomial series in the perturbative parameter, making $e^{\mathcal{S}}$ a unitary transformation. Requiring that $\tilde{\mathcal{H}}^{AB} = 0$ gives a recursive equation for $S_n$, whose terms are nested commutators between the series of $\mathcal{S}$ and $\mathcal{H}$. Similarly, the transformed Hamiltonian is given by a series of nested commutators

$$\tilde{\mathcal{H}} = \sum_{j=0}^{\infty} \frac{1}{j!}\Big[\mathcal{H}, \sum_{n=0}^{\infty} S_n\Big]^{(j)}, \tag{10}$$

---

[4]If both $\mathcal{A}$ and $\mathcal{B}$ are known in advance, fast Fourier transform-based algorithms can reduce this cost to $\sim N \log N$. In our problem, however, the series are constructed recursively and therefore this optimization is impossible.

where the superscript $(j)$ denotes the $j$-th nested commutator $[A, B]^{(j)} = [[A, B]^{(j-1)}, B]$, with $[A, B]^{(0)} = A$ and $[A, B]^{(1)} = AB - BA$. Regardless of the specific implementation, this expression does not meet either of our two requirements:

- The direct computation of the series elements requires $\sim \exp N$ multiplications, and even an optimized one has a $\sim N^2$ scaling.

- Evaluating Eq. (10) contains multiplications by $H_0$.

Alternative parametrizations of the unitary transformation $\mathcal{U}$ require solving unitarity and block diagonalization conditions too, but give rise to a different recursive procedure for the series elements. For example, using hyperbolic functions

$$\mathcal{U} = \cosh \mathcal{G} + \sinh \mathcal{G}, \quad \mathcal{G} = \sum_{i=0}^{\infty} G_i, \tag{11}$$

leads to different recursive expressions for $G_i$ [8], but does not change the algorithm's complexity. On the other hand, using a polynomial series directly

$$\mathcal{U} = \sum_{i=0}^{\infty} U_i, \tag{12}$$

gives rise to another recursive equation for $U_i$ [3, 7, 9, 10]. Still, this choice results in an expression for $\tilde{\mathcal{H}}$ whose terms include products by $H_0$, and therefore requires additional simplifications.

The following three algorithms satisfy both of our requirements while solving a related problem. First, density matrix perturbation theory [11, 12, 29] constructs the density matrix $\rho$ of a perturbed system as a power series with respect to a perturbative parameter:

$$\rho = \sum_{i=0}^{\infty} \rho_i. \tag{13}$$

The elements of the series are found by solving two recursive conditions, $\rho^2 = \rho$ and $[\mathcal{H}, \rho] = 0$, which avoid multiplications by $H_0$ and require a single Cauchy product each. This approach, however, deals with the entire Hilbert space, rather than the low-energy subspace, and does not provide an effective Hamiltonian. Second, the perturbative similarity transform by C. Bloch [2, 13] constructs the effective Hamiltonian in a non-orthogonal basis, which preserves the Hamiltonian spectrum while breaking its hermiticity. Finally, the recursive Schrieffer–Wolff algorithm [24] applies the Schrieffer–Wolff transformation to the output of lower-order iterations, and calculates the effective Hamiltonian at a fixed perturbation strength, rather than as a series. We thus identify the following open question: can we construct an effective Hamiltonian with a linear scaling algorithm that produces compact expressions?

### 3.3 Pymablock's algorithm

The first idea that Pymablock exploits is the recursive evaluation of the operator series, which we illustrate by considering the unitarity condition. Let us separate the transformation $\mathcal{U}$ into an identity and $\mathcal{U}' = \mathcal{W} + \mathcal{V}$:

$$\mathcal{U} = 1 + \mathcal{U}' = 1 + \mathcal{W} + \mathcal{V}, \quad \mathcal{W}^\dagger = \mathcal{W}, \quad \mathcal{V}^\dagger = -\mathcal{V}. \tag{14}$$

We use the unitarity condition $\mathcal{U}^\dagger \mathcal{U} = 1$ by substituting $\mathcal{U}'$ into it:

$$1 = (1 + \mathcal{U}'^\dagger)(1 + \mathcal{U}') = 1 + \mathcal{U}'^\dagger + \mathcal{U}' + \mathcal{U}'^\dagger \mathcal{U}'. \tag{15}$$

This immediately yields

$$\mathcal{W} = \frac{1}{2}(\mathcal{U}'^\dagger + \mathcal{U}') = -\frac{1}{2}\mathcal{U}'^\dagger \mathcal{U}'. \tag{16}$$

Because $\mathcal{U}'$ has no 0-th order term, $(\mathcal{U}'^\dagger \mathcal{U}')_\mathbf{n}$ does not depend on the $\mathbf{n}$-th order of $\mathcal{U}'$ nor $\mathcal{W}$, and therefore Eq. (16) allows to compute $\mathcal{W}$ using the already available lower orders of $\mathcal{U}'$. Alternatively, using Eq. (14) we could define $\mathcal{W}$ as a Taylor series in $\mathcal{V}$:

$$\mathcal{W} = \sqrt{1 + \mathcal{V}^2} - 1 \equiv f(\mathcal{V}) \equiv \sum_n a_n \mathcal{V}^{2n}.$$

A direct computation of all possible products of terms in this expression requires $\sim \exp N$ multiplications. A more efficient approach for evaluating this expression introduces each term in the sum as a new series $\mathcal{A}^{n+1} = \mathcal{A}\mathcal{A}^n$ and reuses the previously computed results. This optimization brings the exponential cost down to $\sim N^2$. However, we see that the Taylor expansion approach is both more complicated and more computationally expensive than the recurrent definition in Eq. (16). Therefore, we use Eq. (16) to efficiently compute $\mathcal{W}$. More generally, a Cauchy product $\mathcal{AB}$ where $\mathcal{A}$ and $\mathcal{B}$ have no 0-th order terms depends on $\mathcal{A}_1, \ldots, \mathcal{A}_{n-1}$ and $\mathcal{B}_1, \ldots, \mathcal{B}_{n-1}$. This makes it possible to use $\mathcal{AB}$ in a recurrence relation, a property that we exploit throughout the algorithm.

To compute $\mathcal{U}'$ we also need to find $\mathcal{V}$, which is defined by the requirement $\tilde{\mathcal{H}}^{AB} = 0$. Additionally, we constrain $\mathcal{V}$ to be block off-diagonal: $\mathcal{V}^{AA} = \mathcal{V}^{BB} = 0$, a choice we make to ensure that the resulting unitary transformation is equivalent to the Schrieffer–Wolff transformation (see section 3.4). In turn, this makes $\mathcal{W}$ block-diagonal and makes the norm of $\mathcal{U}'$ minimal.

The remaining condition for finding a recurrent relation for $\mathcal{U}'$ is that the transformed Hamiltonian

$$\tilde{\mathcal{H}} = \mathcal{U}^\dagger \mathcal{H} \mathcal{U} = \mathcal{H}_D + \mathcal{U}'^\dagger \mathcal{H}_D + \mathcal{H}_D \mathcal{U}' + \mathcal{U}'^\dagger \mathcal{H}_D \mathcal{U}' + \mathcal{U}^\dagger \mathcal{H}'_O \mathcal{U}, \tag{17}$$

is block-diagonal, a condition that determines $\mathcal{V}$. Here we used $\mathcal{U} = 1 + \mathcal{U}'$ and $\mathcal{H} = \mathcal{H}_D + \mathcal{H}'_O$, since $H_0$ is block-diagonal by definition. Because we want to avoid products by $\mathcal{H}_D$, we need to get rid of the terms that contain it by replacing them with an alternative expression. Our strategy is to define an auxiliary operator $\mathcal{X}$ that we can compute without ever multiplying by $\mathcal{H}_D$. Like $\mathcal{U}'$, $\mathcal{X}$ needs to be defined via a recurrence relation, which we determine later. Because Eq. (17) contains $\mathcal{H}_D$ multiplied by $\mathcal{U}'$ from the left and from the right, eliminating $\mathcal{H}_D$ requires moving it to the same side. To achieve this, we choose $\mathcal{X} = \mathcal{Y} + \mathcal{Z}$ to be the commutator between $\mathcal{U}'$ and $\mathcal{H}_D$:

$$\mathcal{X} \equiv [\mathcal{U}', \mathcal{H}_D] = \mathcal{Y} + \mathcal{Z}, \quad \mathcal{Y} \equiv [\mathcal{V}, \mathcal{H}_D] = \mathcal{Y}^\dagger, \quad \mathcal{Z} \equiv [\mathcal{W}, \mathcal{H}_D] = -\mathcal{Z}^\dagger, \tag{18}$$

where $\mathcal{Y}$ is therefore block off-diagonal and $\mathcal{Z}$, block diagonal. We use $\mathcal{H}_D \mathcal{U}' = \mathcal{U}' \mathcal{H}_D - \mathcal{X}$ to move $\mathcal{H}_D$ through to the right and find

$$\begin{aligned}
\tilde{\mathcal{H}} &= \mathcal{H}_D + \mathcal{U}'^\dagger \mathcal{H}_D + (\mathcal{H}_D \mathcal{U}') + \mathcal{U}'^\dagger \mathcal{H}_D \mathcal{U}' + \mathcal{U}^\dagger (\mathcal{H}'_O \mathcal{U}) \\
&= \mathcal{H}_D + \mathcal{U}'^\dagger \mathcal{H}_D + \mathcal{U}' \mathcal{H}_D - \mathcal{X} + \mathcal{U}'^\dagger (\mathcal{U}' \mathcal{H}_D - \mathcal{X}) + \mathcal{U}^\dagger \mathcal{H}'_O \mathcal{U} \\
&= \mathcal{H}_D + (\mathcal{U}'^\dagger + \mathcal{U}' + \mathcal{U}'^\dagger \mathcal{U}') \mathcal{H}_D - \mathcal{X} - \mathcal{U}'^\dagger \mathcal{X} + \mathcal{U}^\dagger \mathcal{H}'_O \mathcal{U} \\
&= \mathcal{H}_D - \mathcal{X} - \mathcal{U}'^\dagger \mathcal{X} + \mathcal{U}^\dagger \mathcal{H}'_O \mathcal{U},
\end{aligned} \tag{19}$$

where the terms multiplied by $\mathcal{H}_D$ cancel according to Eq. (15). The transformed Hamiltonian does not contain multiplications by $\mathcal{H}_D$ anymore, but it does depend on $\mathcal{X}$, an auxiliary operator whose recurrent definition we do not know yet. To find it, we first focus on its anti-Hermitian part, $\mathcal{Z}$. Since recurrence relations are expressions whose right-hand side contains Cauchy products between series, we need to find a way to make a product appear. We do so by using the unitarity condition $\mathcal{U}'^\dagger + \mathcal{U}' = -\mathcal{U}'^\dagger \mathcal{U}'$ to obtain the recursive definition of $\mathcal{Z}$:

$$
\begin{aligned}
\mathcal{Z} &= \frac{1}{2}(\mathcal{X} - \mathcal{X}^\dagger) \\
&= \frac{1}{2}\left[(\mathcal{U}' + \mathcal{U}'^\dagger)\mathcal{H}_D - \mathcal{H}_D(\mathcal{U}' + \mathcal{U}'^\dagger)\right] \\
&= \frac{1}{2}\left[-\mathcal{U}'^\dagger(\mathcal{U}'\mathcal{H}_D - \mathcal{H}_D\mathcal{U}') + (\mathcal{U}'\mathcal{H}_D - \mathcal{H}_D\mathcal{U}')^\dagger\mathcal{U}'\right] \\
&= \frac{1}{2}(-\mathcal{U}'^\dagger\mathcal{X} + \mathcal{X}^\dagger\mathcal{U}').
\end{aligned}
\tag{20}
$$

Similar to computing $W_{\mathbf{n}}$, computing $Z_{\mathbf{n}}$ requires lower-orders of $\mathcal{X}$ and $\mathcal{U}'$. Then, we compute the Hermitian part of $\mathcal{X}$ by requiring that $\tilde{\mathcal{H}}^{AB} = 0$ in the Eq. (19) and find

$$
\mathcal{X}^{AB} = (\mathcal{U}^\dagger \mathcal{H}'_O \mathcal{U} - \mathcal{U}'^\dagger \mathcal{X})^{AB}.
\tag{21}
$$

Once again, despite $\mathcal{X}$ enters the right hand side, because all the terms lack $0^{\text{th}}$ order, this defines a recursive relation for $\mathcal{X}^{AB}$, and therefore $\mathcal{Y}$.

The final part is straightforward: using $\mathcal{H}_D = H_0 + \mathcal{H}'_D$ and the definition of $\mathcal{Y}$ in Eq. (18) fixes $\mathcal{V}$ as a solution of:

$$
\mathcal{V}^{AB} H_0^{BB} - H_0^{AA} \mathcal{V}^{AB} = \mathcal{Y}^{AB} - [\mathcal{V}, \mathcal{H}'_D]^{AB},
\tag{22}
$$

a Sylvester's equation, which we only need to solve once for every new order. In the eigenbasis of $H_0$, the solution of Sylvester's equation is $V_{\mathbf{n},ij}^{AB} = (\mathcal{Y} - [\mathcal{V}, \mathcal{H}'_D])_{\mathbf{n},ij}^{AB}/(E_i - E_j)$, where $E_i$ are the eigenvalues of $H_0$.

We now have the complete algorithm:

1. Define series $\mathcal{U}'$ and $\mathcal{X}$ and make use of their block structure and Hermiticity.

2. To define the diagonal blocks of $\mathcal{U}'$, use $\mathcal{W} = -\mathcal{U}'^\dagger\mathcal{U}'/2$.

3. To find the off-diagonal blocks of $\mathcal{U}'$, solve Sylvester's equation
   $\mathcal{V}^{AB} H_0^{BB} - H_0^{AA} \mathcal{V}^{AB} = \mathcal{Y}^{AB} - [\mathcal{V}, \mathcal{H}'_D]^{AB}$. This requires $\mathcal{X}$.

4. To find the diagonal blocks of $\mathcal{X}$, define $\mathcal{Z} = (-\mathcal{U}'^\dagger\mathcal{X} + \mathcal{X}^\dagger\mathcal{U}')/2$.

5. For the off-diagonal blocks of $\mathcal{X}$, use $\mathcal{Y}^{AB} = (-\mathcal{U}'^\dagger\mathcal{X} + \mathcal{U}^\dagger\mathcal{H}'\mathcal{U})^{AB}$.

6. Compute the effective Hamiltonian as $\tilde{\mathcal{H}} = \mathcal{H}_D - \mathcal{X} - \mathcal{U}'^\dagger\mathcal{X} + \mathcal{U}^\dagger\mathcal{H}'_O\mathcal{U}$.

### 3.4 Equivalence to Schrieffer–Wolff transformation

Pymablock's algorithm and the Schrieffer–Wolff transformation both find a unitary transformation $\mathcal{U}$ such that $\tilde{\mathcal{H}}^{AB} = 0$. They are therefore equivalent up to a gauge choice in each subspace, $A$ and $B$. We establish the equivalence between the two by demonstrating that this gauge choice is the same for both algorithms. The Schrieffer–Wolff transformation uses $\mathcal{U} = \exp\mathcal{S}$, where $\mathcal{S} = -\mathcal{S}^\dagger$ and $\mathcal{S}^{AA} = \mathcal{S}^{BB} = 0$, this restriction makes the result unique [2]. On the other hand, our algorithm produces the unique block-diagonalizing

transformation with a block structure $\mathcal{U}_{AA} = \mathcal{U}_{AA}^\dagger$, $\mathcal{U}_{BB} = \mathcal{U}_{BB}^\dagger$ and $\mathcal{U}_{AB} = -\mathcal{U}_{BA}^\dagger$. The uniqueness is a consequence of the construction of the algorithm, where calculating every order gives a unique solution satisfying these conditions. To see that the two solutions are identical, we expand the Taylor series of $\exp \mathcal{S}$. Every even order gives a Hermitian, block-diagonal matrix, while every odd order gives an anti-Hermitian block off-diagonal matrix, showing that $\exp \mathcal{S}$ has the same structure as $\mathcal{U}$ above. The reverse statement about the structure of $\log \mathcal{U}$ can be seen similarly, using the Taylor series of the logarithm around 1. Using a series expansion is justified by the perturbative nature of the result, meaning that $\mathcal{S}$ is close to 0 and $\mathcal{U}$ is close to 1. Because of the uniqueness of both results, we find that $\exp \mathcal{S}$ from conventional Schrieffer–Wolff transformation is identical to $\mathcal{U}$ found by our algorithm, which remains true if both power series are truncated at a finite order.

### 3.5   Extra optimization: common subexpression elimination

While the algorithm of Sec. 3.3 satisfies our requirements, we improve it further by reusing products that are needed in several places, such that the total number of matrix multiplications is reduced. Firstly, we rewrite the expressions for $\mathcal{Z}$ in Eq. (20) and $\tilde{\mathcal{H}}$ in Eq. (19) by utilizing the Hermitian conjugate of $\mathcal{U}'^\dagger \mathcal{X}$ without recomputing it:

$$\mathcal{Z} = \frac{1}{2}\left[(-\mathcal{U}'^\dagger \mathcal{X}) - \text{h.c.}\right],$$
$$\tilde{\mathcal{H}} = \mathcal{H}_D + \mathcal{U}^\dagger \mathcal{H}_O' \mathcal{U} - (\mathcal{U}'^\dagger \mathcal{X} + \text{h.c.})/2,$$

where h.c. is the Hermitian conjugate, and $\mathcal{X}$ drops out from the diagonal blocks of $\tilde{\mathcal{H}}$ because the diagonal blocks of $\mathcal{X}$ are anti-Hermitian. Additionally, we reuse the repeated $\mathcal{A} \equiv \mathcal{H}_O' \mathcal{U}'$ in

$$\mathcal{U}^\dagger \mathcal{H}_O' \mathcal{U} = \mathcal{H}_O' + \mathcal{A} + \mathcal{A}^\dagger + \mathcal{U}'^\dagger \mathcal{A}. \tag{23}$$

Next, we observe that some products from the $\mathcal{U}^\dagger \mathcal{H}_O \mathcal{U}$ term appear both in $\mathcal{X}$ in Eq. (21) and in $\tilde{\mathcal{H}}$ (19). To avoid recomputing these products, we introduce $\mathcal{B} = \mathcal{X} - \mathcal{H}_O' - \mathcal{A}$ and define the recursive algorithm using $\mathcal{B}$ instead of $\mathcal{X}$. With this definition, we compute the off-diagonal blocks of $\mathcal{B}$ as:

$$\begin{aligned}
\mathcal{B}^{AB,BA} &= \left[\mathcal{X} - \mathcal{H}_O' - \mathcal{A}\right]^{AB,BA} \\
&= \left[\mathcal{A}^\dagger + \mathcal{U}'^\dagger \mathcal{A} - \mathcal{U}'^\dagger \mathcal{X}\right]^{AB,BA} \\
&= \left[\mathcal{U}'^\dagger \mathcal{H}_O' + \mathcal{U}'^\dagger \mathcal{A} - \mathcal{U}'^\dagger \mathcal{X}\right]^{AB,BA} \\
&= -(\mathcal{U}'^\dagger \mathcal{B})^{AB,BA},
\end{aligned} \tag{24}$$

where we also used Eq. (21) and the definition of $\mathcal{A}$. The diagonal blocks of $\mathcal{B}$, on the other hand, are given by

$$\begin{aligned}
\mathcal{B}^{AA,BB} &= \left[\mathcal{X} - \mathcal{H}_O' - \mathcal{A}\right]^{AA,BB} \\
&= \left[\frac{1}{2}[(-\mathcal{U}'^\dagger \mathcal{X}) - \text{h.c.}] - \mathcal{A}\right]^{AA,BB} \\
&= \left[\frac{1}{2}[(-\mathcal{U}'^\dagger[\mathcal{X} - \mathcal{H}_O' - \mathcal{A}]) - \text{h.c.}] - \frac{1}{2}[\mathcal{A}^\dagger + \mathcal{A}] + \frac{1}{2}[(-\mathcal{U}'^\dagger \mathcal{A}) - \text{h.c.}]\right]^{AA,BB}, \\
&= \left[\frac{1}{2}[(-\mathcal{U}'^\dagger \mathcal{B}) - \text{h.c.}] - \frac{1}{2}[\mathcal{A}^\dagger + \text{h.c.}]\right]^{AA,BB},
\end{aligned} \tag{25}$$

where we used Eq. (20) and that $\mathcal{U}'^\dagger \mathcal{A}$ is Hermitian. Using $\mathcal{B}$ changes the relation for $\mathcal{V}^{AB}$ in Eq. (22) to

$$\mathcal{V}^{AB} H_0^{BB} - H_0^{AA} \mathcal{V}^{AB} = \left( \mathcal{B} - \mathcal{H}' - \mathcal{A} - [\mathcal{V}, \mathcal{H}'_D] \right)^{AB}. \qquad (26)$$

Finally, we combine Eq. (19), Eq. (23), Eq. (25) and Eq. (24) to obtain the final expression for the effective Hamiltonian:

$$\tilde{\mathcal{H}}_D = \mathcal{H}_D + (\mathcal{A} + \text{h.c.})/2 - (\mathcal{U}^\dagger \mathcal{B} + \text{h.c.})/2. \qquad (27)$$

Together with the series $\mathcal{U}'$ in Eqs. (16,26), $\mathcal{A} = \mathcal{H}'_O \mathcal{U}'$, and $\mathcal{B}$ in Eqs. (25,24), this equation defines the optimized algorithm.

## 4 Implementation

### 4.1 The data structure for block operator series

The optimized algorithm from the previous section requires constructing 14 operator series, whose elements are computed using a collection of recurrence relations. This warrants defining a specialized data structure suitable for this task that represents a multidimensional series of operators. Because the recurrent relations are block-wise, the data structure needs to keep track of separate blocks. In order to support varied use cases, the actual representation of the operators needs to be flexible: the block may be dense arrays, sparse matrices, symbolic expressions, or more generally any object that defines addition and multiplication. Finally, the series needs to be queryable by order and block, so that it supports a block-wise multivariate Cauchy product—the main operation in the algorithm.

The most straightforward way to implement a perturbation theory calculation is to write a function that has the desired order as an argument, computes the series up to that order, and returns the result. This makes it hard to reuse already computed terms for a new computation, and becomes complicated to implement in the multidimensional case when different orders in different perturbations are needed. We find that a recursive approach addresses these issues: within this paradigm, each series needs to define how its entries depend on lower-order terms.

To address these requirements, we define a `BlockSeries` Python class and use it to represent the series of $\mathcal{U}$, $\mathcal{H}$, and $\tilde{\mathcal{H}}$, as well as the intermediate series used to define the algorithm. The objects of this class are equipped with a function to compute their elements and it stores the already computed results in a dictionary. Storing the results for reuse is necessary to optimize the evaluation of higher order terms and it allows to request additional orders without restarting the computation. For example, the definition of the `BlockSeries` for $\tilde{\mathcal{H}}$ has the following form:

```
H_tilde = BlockSeries(
    shape=(2, 2),  # 2x2 block matrix
    n_infinite=n,  # number of perturbative parameters
    eval=compute_H_tilde,  # function to compute the elements
    name="H_tilde",
    dimension_names=("lambda", ...),  # parameter names
)
```

Here `compute_H_tilde` is a function implementing Eq. (27) by querying other series objects. Calling `H_tilde[0, 0, 2]`, the second order perturbation $\sim \lambda^2$ of the $AA$ block, then does the following:

1. Evaluates `compute_H_tilde(0, 0, 2)` if it is not already computed.

2. Stores the evaluation result in a dictionary.

3. Returns the result.

To conveniently access multiple orders at once, we implement NumPy array indexing so that `H_tilde[0, 0, :3]` returns a NumPy masked array array with the orders $\sim \lambda^0$ , $\sim \lambda^1$, and $\sim \lambda^2$ of the $AA$ block. The masking allows to support a common use case where some orders of a series are zero, so that they are omitted from the computations. We expect that the `BlockSeries` data structure is suitable to represent a broad class of perturbative calculations, and we plan to extend it to support more advanced features in the future.

We utilize `BlockSeries` to implement multiple other optimizations. For example, we exploit Hermiticity when computing the Cauchy product of $U'^{\dagger}U'$ in Eq. (16), by only evaluating half of the matrix products, and then complex conjugate the result to obtain the rest. Similarly, for Hermitian and anti-Hermitian series, like the off-diagonal blocks of $\mathcal{U}'$, we only compute the $AB$ blocks, and use the conjugate transpose to obtain the $BA$ blocks. This approach should also allow us to implement efficient handling of symmetry-constrained Hamiltonians, where some blocks either vanish or are equal to other blocks due to a symmetry. Moreover, using `BlockSeries` with custom objects yields additional information about the algorithm and accommodates its further development. Specifically, we have used a custom object with a counter to measure the algorithm complexity (see also Sec. 5) and to determine which results are only used once so that they can be immediately discarded from storage.

## 4.2 The implicit method for large sparse Hamiltonians

A distinguishing feature of Pymablock is its ability to handle large sparse Hamiltonians, that are too costly to diagonalize, as illustrated in Sec. 2.3. Specifically, we consider the situations when the size $N_A$ of the $A$ subspace is small compared to the entire Hilbert space, so that obtaining the basis $\Psi_A$ of the $A$ subspace is feasible using sparse diagonalization. The projector on the $A$ subspace $P_A = \Psi_A^{\dagger} \Psi_A$ is then a low-rank matrix, a property that we exploit to avoid constructing the $B$ subspace explicitly. Furthermore, the solution of Sylvester's equation in Eq. 22 amounts to multiplying $N_A$ large vectors, rows of $Y_{\mathbf{n}}^{AB}$, by the energy denominators $E_i - E_j$, where $E_i$ are the $N_A$ eigenvalues of the $A$ subspace provided by sparse diagonalization.

The key tool to solve this problem is the projector approach introduced in Ref. [30], which introduces an equivalent extended Hamiltonian using the projector $P_B = 1 - P_A$ onto the $B$ subspace:

$$\bar{\mathcal{H}} = \begin{pmatrix} \Psi_A^{\dagger}\mathcal{H}\Psi_A & \Psi_A^{\dagger}\mathcal{H}P_B \\ P_B\mathcal{H}\Psi_A & P_B\mathcal{H}P_B \end{pmatrix}. \tag{28}$$

In other words, the subspace $\bar{A}$ is written in the basis of $\Psi_A$, while the basis of the $\bar{B}$ subspace is the same as the original complete basis of $\mathcal{H}$ to preserve its sparsity. We also project out the $A$-degrees of freedom from the $\bar{B}$ subspace to avoid duplicate solutions in $\bar{\mathcal{H}}$, which introduces $N_A$ eigenvectors with zero eigenvalues. Introducing $\bar{\mathcal{H}}$ allows to multiply by operators of a form $P_B H_{\mathbf{n}} P_B$ efficiently by using the low-rank structure of $P_A$. In the code we represent the $\bar{B}\bar{B}$ operators as `LinearOperator` objects from the SciPy package [28], enabled by the ability of the `BlockSeries` to store arbitrary objects. Storing the $\bar{A}\bar{A}$ and $\bar{A}\bar{B}$ blocks as dense matrices—efficient because these are small and dense—finishes the implementation of the Hamiltonian.

To solve the Sylvester's equation we write it for every row of $V_{\mathbf{n}}^{\bar{A}\bar{B}}$ separately:

$$V_{\mathbf{n},ij}^{\bar{A}\bar{B}}(E_i - H_0) = Y_{\mathbf{n},j}^{\bar{A}\bar{B}} \tag{29}$$

This equation has a solution despite $E_i - H_0$ not being invertible because $Y_{\mathbf{n}}^{\bar{A}\bar{B}} P_A = 0$. We solve this equation using the MUMPS sparse solver [31, 32], which prepares an efficient sparse LU-decomposition of $E_i - H_0$, or the KPM approximation of the Green's function [33]. Both methods work on sparse Hamiltonians with millions of degrees of freedom.

# 5    Benchmark

To the best of our knowledge, there are no other packages implementing arbitrary order quasi-degenerate perturbation theory. Literature references provide explicit expressions for the effective Hamiltonian up to fourth order, together with the procedure for obtaining higher order expressions [34]. Because the full reference expressions are lengthy, we do not provide them, but for example at 4-th order the effective Hamiltonian is a sum of several expressions of the form

$$\sum_{m''m'''l} \frac{H'_{mm''} H'_{m''m'''} H'_{m'''l} H'_{lm'}}{(E_{m''} - E_l)(E_{m'''} - E_l)(E_m - E_l)}, \tag{30}$$

where the $m$-indices label states from the $A$-subspace and $l$-indices label the states from the $B$-subspace. More generally, at $n$-th order each term is a product of $n$ matrix elements of the Hamiltonian and $n - 1$ energy denominators. Directly carrying out the summation over all the states requires $\mathcal{O}(N_A^2 N_B^{n-1})$ operations, where $N_A$ and $N_B$ are the number of states in the two subspaces. In other words, the direct computation scales worse than a matrix product with the problem size. Formulating Eq. (30) as $n - 1$ matrix products combined with $n - 1$ solutions of Sylvester's equation, brings this complexity down to $\mathcal{O}((n - 1) \times N_A N_B^2)$. This optimization, together with the hermiticity of the sum, allows us to evaluate the reference expressions for the effective Hamiltonian for 2-nd, 3-rd, and 4-th order using 1, 4, and 27 matrix products, respectively. Pymablock's algorithm utilizes 1, 3, and 14, matrix products to obtain the same orders of the effective Hamiltonian. Its advantage becomes even more pronounced at higher orders due to the exponential growth of the number of terms in the reference expressions. While finding the optimized implementation from the reference expressions is possible for the 3-rd order, we expect it to be extremely challenging for the 4-th order, and essentially impossible to do manually for higher orders. Moreover, because the `BlockSeries` class tracks absent terms, in practice the number of matrix products depends on the sparsity of the block structure of the perturbation, as shown in Fig. 4.

The efficiency of Pymablock becomes especially apparent when applied to sparse numerical problems, similar to Sec. 2.3. We demonstrate the performance of the implicit method by using it to compute the low-energy spectrum of a large tight-binding model, and comparing Pymablock's time cost to that of sparse diagonalization. We define a 2D square lattice of $52 \times 52$ sites with nearest-neighbor hopping and a random onsite potential $\mu(\mathbf{r})$. The perturbation $\delta\mu(\mathbf{r})$ interpolates between two different disorder realizations. For the sake of an illustration, we choose the system's parameters such that the dispersion of the lowest few levels with $\delta\mu$ features avoided crossings and an overall nonlinear shape, whose details are not relevant. Similar to Sec. 2.3, constructing the effective Hamiltonian involves three steps. First, we compute the 10 lowest states of the unperturbed Hamiltonian using sparse diagonalization. Second, `block_diagonalize` computes a sparse LU decomposition of the Hamiltonian at each of the 10 eigenenergies. Third, we compute corrections $\tilde{H}_1$, $\tilde{H}_2$, and $\tilde{H}_3$ to the effective Hamiltonian, each being a $10 \times 10$ matrix. Each of these steps is a one-time cost, see Fig. 5. Finally, to compare the perturbative calculation to sparse

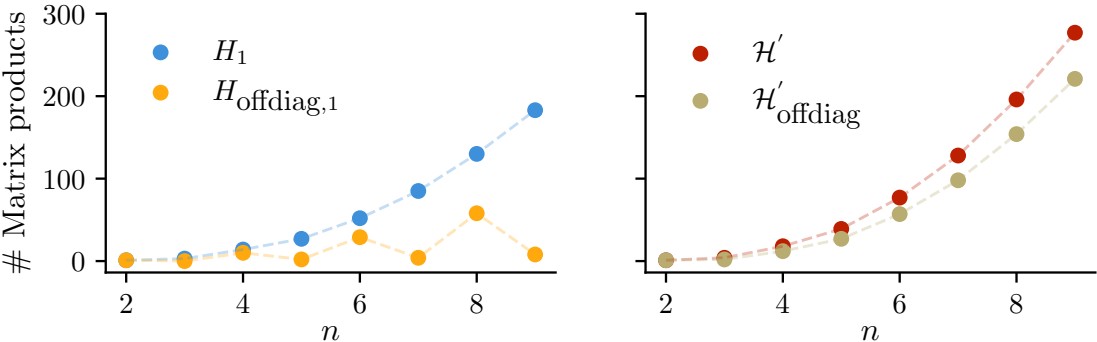

Figure 4: Matrix products required to compute $\tilde{H}_n^{AA}$ for a dense and block off-diagonal first-order perturbation (left) and a dense and block off-diagonal perturbative series with terms of all orders present (right).

diagonalization, we construct the effective Hamiltonian $\tilde{H} = H_0 + \delta\mu\tilde{H}_1 + \delta\mu^2\tilde{H}_2 + \delta\mu^3\tilde{H}_3$ and diagonalize it to obtain the low-energy spectrum for a range of $\delta\mu$. This has a negligible cost compared to constructing the series. The comparison is shown in Fig. 5. We observe that while the second order results are already very close to the exact spectrum, the third order corrections fully reproduce the sparse diagonalization. At the same time, the entire cost of computing the perturbative band structure for a range of $\delta\mu$ is lower than computing a single additional sparse diagonalization.

# 6 Conclusion

We developed an algorithm for constructing an effective Hamiltonian that combines advantages of different perturbative expansions. The main building block of our approach is a set of recurrence relations that define several series that depend on each other and combine into the effective Hamiltonian. Our algorithm constructs the same effective Hamiltonians as the Schrieffer–Wolff transformation [1], while keeping the linear scaling per extra order similar to the density matrix perturbation theory [11,12] or the non-orthogonal perturbation theory [13]. Its expressions minimize the number of matrix multiplications per order, making it appealing both for symbolic and numerical computations.

We provide a Python implementation of the algorithm in the Pymablock package [35]. The package is thoroughly tested (94% test coverage as of version 2.0), becoming a reliable tool for constructing effective Hamiltonians that combine multiple perturbations to high orders. The core of the Pymablock interface is the `BlockSeries` class that handles arbitrary objects as long as they support algebraic operations. This enables Pymablock's construction of effective models for large tight-binding models using its implicit method. It also allows Pymablock to solve both symbolic and numerical problems in diverse physical settings, and potentially to incorporate it into existing packages, such as scqubits [22], QuTiP [36,37], or dft2kp [38].

Beyond the Schrieffer–Wolff transformation, the Pymablock package provides a foundation for defining other perturbative expansions. We anticipate extending it to time-dependent problems, where the different regimes of the time-dependent drive modify the recurrence relations that need to be solved. Applying the same framework to problems with weak position dependence would allow to construct a nonlinear response theory of quantum materials. Finally, we expect that in the many-particle context the same frame-

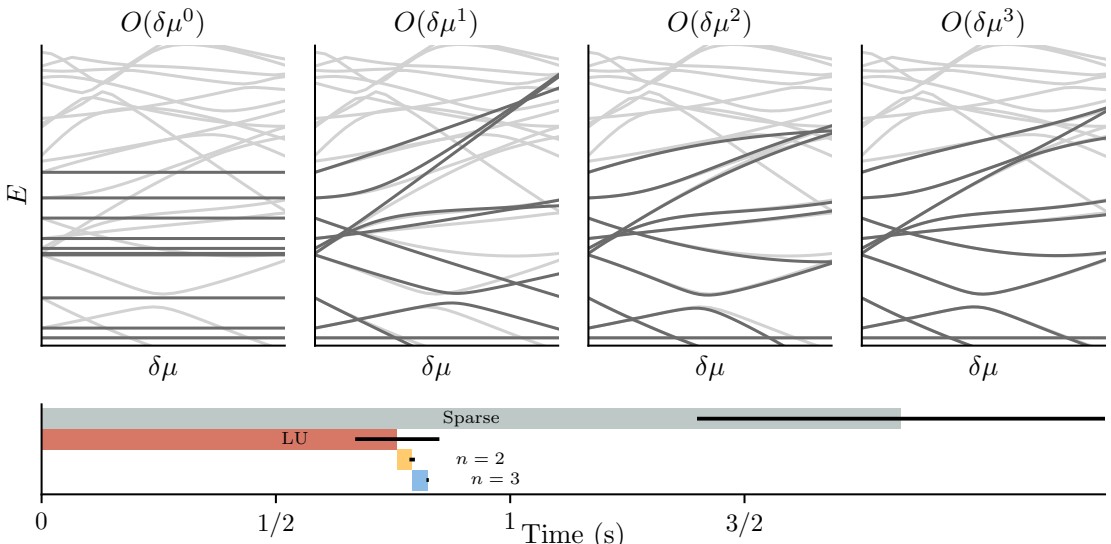

Figure 5: Top panels: band structure of the perturbative effective Hamiltonian (black) of a tight-binding model compared to exact sparse diagonalization (gray). Bottom panel: a comparison of the Pymablock's time cost with sparse diagonalization. Most of the time is spent in the LU decomposition of the Hamiltonian (red). The entire cost of the implicit method is lower than a single sparse diagonalization (gray). The operations of negligible cost are not shown. The bars length corresponds to the average time cost over 40 runs, and the error bars show the standard deviation.

work supports implementing different flavors of diagrammatic expansions.

## Acknowledgements

We thank Valla Fatemi and Antonio Manesco for feedback on the manuscript.

## Data availability

The code used to produce the reported results is available on Zenodo [35].

**Author contributions** A. R. A. had the initial idea and oversaw the project. All authors developed the algorithm. I. A. D., S. M., H. K. K, and A. R. A. wrote the package. I. A. D. and A. R. A. wrote the paper.

**Funding information** This research was supported by the Netherlands Organization for Scientific Research (NWO/OCW) as part of the Frontiers of Nanoscience program, a NWO VIDI grant 016.Vidi.189.180, and OCENW.GROOT.2019.004. D.V. acknowledges funding from the Deutsche Forschungsgemeinschaft (DFG, German Research Foundation) under Germany's Excellence Strategy through the Würzburg-Dresden Cluster of Excellence on Complexity and Topology in Quantum Matter – ct.qmat (EXC 2147, project-ids 390858490 and 392019).

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
