# Peer review of "Pymablock: an algorithm and a package for quasi-degenerate perturbation theory"

_SciPost Physics Codebases, doi:SciPost Phys. Codebases 50-r2.1 (2025) , SciPost Phys. Codebases 50 (2025)_

## Round 1 · Referee Report · Anonymous (Referee 1) · 2024-7-30

Report

Report for “Pymablock: an algorithm and a package for quasi-degenerate perturbation theory” by Day et al.

The Python package “Pymablock” implements a unitary transformation for the block diagonalization of a Hermitian matrix using a perturbative approach. This method allows for symbolic evaluation and uses a recursive approach for the operator series expressions, equivalent to the Schrieffer-Wolff transformation, with the primary step requiring the solution of a Sylvester equation for each order. The package’s capabilities are demonstrated through examples including a k.p model for AB-stacked Graphene, a transmon qubit coupled to a resonator, and a double quantum dot.

I have put the package to the test by investigating a bosonic Hamiltonian with a quartic perturbation, see Eq. (11) in Phys. Rev. B 86, 125113 (2012). First observation: Pymablock cannot directly work with a Hamiltonian defined in second quantization. Indeed, as the authors themselves do it in 2.2 I had to translate the terms into matrix form and truncate the Hilbert space in order for the approach to work. If I chose to block diagonalize the ground state (A) end decouple it from all the other states (B) the approach worked well and provides the correct series expansion. For the first excited state the first order correction (although trivial) was also correct. The second order correction yielded a value of $-30x^2/\omega$ which first confused me, since the corresponding bilinear operator b^dagger b has a series expansion of $-24x^2/\omega$, see table I in above reference. But it is explained by the fact, that Pymablock is not diagonalizing the ground state at this point anymore, thus one would need to also subtract the value $-6x^2/\omega$ from the ground state energy to make connection to previous results. Thus the package works but the drawback of not blockdiagonalizing several blocks becomes apparent.

SciPost Physics Codebase requires the following for acceptance:
1. Benchmarking tests must be provided.
I think this criteria is fulfilled in section 5, but only marginally so. It indeed seems like there are no open source packages available that implement high order series expansion methods.
2. At least one example application must be presented in detail
This criteria is perfectly fulfilled, as the authors discuss 3 examples in their paper.
3. High-level programming standards must be followed throughout the source code
The source code is well documented and from my review seems to be in good condition.
4. The userguide must properly contextualize the software, describe the logic of its workings and highlight its added value as compared to existing software
Also this criteria is fulfilled.
5. The software must address a demonstrable need for the scientific community
This is the biggest issue I have with the paper, see comment below.
6. The documentation must be complete, including detailed instructions on downloading, installing and running the software
This criteria is fulfilled, the package worked out of the box after installing with conda/pip.

Overall I think the packages provides a good first start for a python package aiming at perturbatively decoupling Hamiltonians. However the paper is missing a huge portion of literature that aims exactly at the same issue: continuous unitary transformations (CUT) are a tool that provides effective models based on perturbative expansion. A good overview (although a bit outdated) is given in sections I and II of Phys. Rev. B 86, 125113 (2012). Having in mind the capabilities of these approaches the main drawbacks I see with pymablock are:

1) Lack of simultaneous Block-Diagonalization beyond 2 blocks.
2) Lack of support for second quantized Hamiltonians (i.e. true expansion in terms of prefactors for the operators instead of truncated Matrix representation.

I still think that due to the missing open source implementations of above methods, pymablock fills this gap.

Minor point:
So far only two subspaces are block diagonalized, hence Fig 1 is misleading, as it looks like a block diagonalization of more than 2 blocks!

Therefore my recommendation is to publish, after the authors add a more comprehensive literature review on above mention methods in order to derive perturbative expansions for effective models. In the long-term (for future releases) I recommend the authors to address the points about simultaneous Block-Diagonalization and support for second quantized Hamiltonians.

Requested changes

Add comprehensive literature review on CUT methods for effective perturbative Hamiltonians

Recommendation

Ask for minor revision

---

## Round 2 · Author Response

We also took seriously the referee's observation that Pymablock deals with only two subspaces and the limited support for second quantized Hamiltonians.
In the updated version 2.1 of the package we have developed and implemented multi-block perturbation theory, an approach we now demonstrate in the CQED example and new online tutorials.
Our derivation of the algorithm now also fully relies on the operator formalism, and therefore directly applies to a broad class of problems.
As requested by the referee, we have included the application of CUT methods to perturbation theory in Sec. 3.2.

---

## Round 2 · List of Changes

Below we list the significant changes in the manuscript, and separately we provide a redlined pdf.
- Generalized Pymablock to multiblock and selective diagonalization. Included this in 2.2, 2.4, and 3.1
- Included a description and review of the Continuous Unitary Transformation approach (CUT) in 1 and 3.2
- Described new code generation feature of the package to support the implementation of different algorithms in 4.3
The redlined pdf with the differences can be found in https://surfdrive.surf.nl/files/index.php/s/rzI7bNljOLdDHN3

---

## Editorial Decision

published